# Ag-Ion-Based Transparent Threshold Switching Selector with Filament-Size-Dependent Rectifying Behavior

**DOI:** 10.3390/mi13111874

**Published:** 2022-10-31

**Authors:** Jongseon Seo, Geonhui Han, Hyejin Kim, Daeseok Lee

**Affiliations:** Department of Electronic Materials Engineering, Kwangwoon University, Seoul 01897, Korea

**Keywords:** transparent device, memory switching device, threshold switching device, self-rectifying characteristics

## Abstract

A metal–insulator–metal-structured Ag-filament-based transparent threshold switch is developed as a selector device for a crossbar array, which can lead to high-density integration of advanced memory devices. Both threshold switching and rectifying behavior were achieved based on sensitive control of the filament size. Conduction mechanism analyses demonstrated that the rectifying behavior resulted from the Schottky barrier at the interface. From the threshold switching, including the rectifying behavior, the available crossbar array size is 10${}^{5}$-times larger.

## 1. Introduction

A crossbar array structure was proposed for the high-density integration of advanced memory devices [1,2,3,4,5,6,7,8]. However, the sneak path current of the crossbar array structure can result in information misreading and higher power consumption [9]. Therefore, to overcome these obstacles, various studies have been conducted, such as selector devices and complementary resistive switching [10,11,12]. In particular, a selector device, which can lead to a one-selector and one-memory (1S1M) structure, is one of the most promising structures. Thus, various selector devices have been extensively studied, such as the Ovonic threshold switch, metal–insulator transition selector, and metal (Ag or Cu) filament threshold switch [13,14,15]. Even though selector devices have been widely developed, transparent selector devices that can be utilized for display and transparent electronic goods have not yet been sufficiently researched. In this study, we developed an Ag-filament-based transparent threshold switch.

## 2. Materials and Methods

To develop the Ag-filament-based transparent threshold switch, all layers were fabricated as thin films using a sensitively controlled sputtering process. In addition, current compliance-dependent characteristics such as the threshold switch and memory switch were analyzed. In particular, a rectifying characteristic under a negative bias was observed by controlling the current compliance.

Figure 1a shows the developed Ag-filament-based transparent threshold switch. The 25 nm-thick WO_3_ layer was deposited on an ITO layer. Subsequently, a 2 nm-thick Ag layer and 35 nm-thick ITO top electrodes were patterned into 300 μm × 300 μm squares. Figure 1b shows the cross-sectional transmission electron microscopy (TEM) image of the developed device; the TEM result shows that each layer was well formed. The optical characteristic transmittance in the visible light band of 400 to 750 nm is shown in Figure 1c along with a photograph of the fabricated device. Compared to bare glass, the fabricated device exhibited approximately 75% transmittance. These results demonstrated that the Ag-filament-based transparent threshold switch was successfully developed.

## 3. Results

Figure 2a shows the direct current versus voltage (I–V) characteristics for different current compliance values. Specifically, the memory switches are exhibited at 300 μA and 100 μA current compliance values (red curves in Figure 2a). A positive voltage sweep from 0 V to 3 V formed an Ag-ion-based filament, and the initial high-resistance state (HRS) changed to a low-resistance state (LRS). Subsequently, when the negative voltage was swept from 0 V to −5 V, the resistance of the device changed from LRS to HRS as the Ag filament ruptured. Consequently, the counterclockwise I–V characteristic was measured. A thicker and more stable filament is formed when the current compliance is higher. Thus, at a higher current compliance (300 μA), the filament cannot be ruptured under an applied negative bias [16].

In addition, the 1S1M characteristics and threshold switch were observed at 50 μA and under 10 μA current compliance values, respectively. Under the 10 μA and 2 μA current compliance, relatively thin and unstable filaments were formed, which resulted in a threshold switch. The filament can spontaneously rupture without bias because of its instability. Therefore, counterclockwise (clockwise) I–V was observed under a positive (negative) bias. It should be noted that rectifying behaviors were also detected at 10 μA and 2 μA current compliance values.

In contrast to the threshold switch cases (at 10 μA and 5 μA current compliance), at 50 μA current compliance, a counterclockwise I–V curve including the suppressed current in the low negative voltage region was achieved. This can be considered as a combination of memory and threshold switch behaviors [17].

To analyze the current compliance dependence, the filament size at each compliance current was derived, as shown in Figure 2b,c. Figure 2b shows the resistance in the LRS at various current compliance values. We confirmed that the LRS resistance decreased linearly as the current compliance increased. This means that the Ag filament diameter (D) increases as the current compliance increases, according to Equation (Equation 1). We then calculated the diameter of the Ag filament according to the current compliance using the following equations:(1)$$R_{LRS}=\frac{\rho_{filament}\cdot L}{\pi \cdot r^{2}}$$
(2)$$D=2r=2\sqrt{\frac{\rho_{filament}\cdot L}{\pi \cdot R_{LRS}}}$$
where *L* is the length (thickness of the WO_3_ solid electrolyte, approximately 25 nm), $\rho_{filament}$ is the resistivity of bulk Ag (approximately 1.6 × 10${}^{-8}\Omega \cdot m$), *r* is the radius, and *D* is the diameter of the Ag filament [18,19]. Based on this equation, the *D* values for various current compliance values are plotted as shown in Figure 2c.

Figure 3 shows a schematic of the operation mechanisms according to various current compliances (low current compliance for the 2 and 10 μA cases, medium current compliance for the 50 μA case, and high current compliance for the 100 and 300 μA cases) based on the results. The blue filament represents the selector characteristics with a thin filament, and the red filament represents the memory characteristics with a thick filament. In particular, the selector filament exhibits unstable characteristics even in the on state because of the decreased diameter of the filament. As the diameter of the filament decreases, the metal bonds and the force to bind metal atoms to each other also decrease. Therefore, when the voltage bias was removed, it became easier for the Ag atoms composing the filament to be moved by diffusion through the defect sites, which is schematically described by the red arrows in Figure 3a [20].

Figure 4a,b show the I–V curves in the off state of 2 μA and 10 μA current compliance, respectively. Even though both cases exhibited rectifying behavior, a more obvious rectifying behavior was observed at 2 μA current compliance. Considering that the half-bias scheme that employs full-bias (V) and half-bias (V/2) for selected and unselected cells, respectively, can be utilized in the crossbar array, V and V/2 of the threshold switch can be defined as shown in Figure 4a,b. Based on the defined V and V/2, the rectifying ratios (R.R) were determined as 274 and 11.3 for the 2 μA and 10 μA current compliance cases, respectively, according to the equation given below.
(3)$$Rectifying\;Ratio\;\left(R.R\right)=\frac{Current\;at\;V}{Current\;at\;V/2}$$

To analyze these results in detail, for the 2 μA and 10 μA current compliance, a conduction mechanism analysis was performed by fitting the I–V curves. In the positive-bias region, both cases exhibited the same conduction mechanisms. First, thermionic field emission, which includes the coexistence of Schottky emission and tunneling, was observed in the low-electric-field region. Then, in the high-electric-field region, Poole–Frenkel conduction was observed.

Unlike the positive-bias region, in the negative-bias region, the devices exhibited different conduction mechanisms. At the 10 μA current compliance, Schottky emission and direct tunneling were dominant in the low and high electric fields, respectively. However, at 2 μA current compliance, only Schottky emission was observed under a negative bias. Considering that Schottky emission is caused by the presence of a Schottky barrier, we can assume that a Schottky barrier can be formed between the WO_3_ and ITO interface [21,22].

To confirm this, we fabricated a symmetrically structured device without an Ag layer (ITO/WO_3_/ITO), as shown in Figure 5a. The device exhibited bidirectional rectifying behavior, which fit well with the Schottky emission equation. This result clearly demonstrated the presence of a Schottky barrier between the WO_3_ and ITO layers.

Based on this result, for the 2 μA and 10 μA current compliance values, the Schottky barrier height and remaining gap between the filament and electrode were derived, as shown in Figure 5b,c. From the Schottky emission fit curves in the low-negative-electric-field region, intercepts and slopes were extracted. The Schottky barrier height and the remaining gap were derived using the following equations.
(4)$$\left|Intercept\right|\propto \varphi_{B}$$
(5)$$Slope\propto \frac{1}{\sqrt{\varepsilon_{\iota }\times d}}$$
where $\varphi_{B}$, $\varepsilon_{\iota }$, and d are the barrier height, relative permittivity, and Schottky barrier width (remaining gap), respectively [22]. The height of the Schottky barrier is proportional to the absolute value of the intercept, while the Schottky barrier width is inversely proportional to the slope. At 10 μA current compliance, the Schottky barrier height and width were relatively smaller than those in the 2 μA current compliance case, as shown in Figure 5d. These results imply that direct tunneling of the 10 μA current compliance case (under a high-negative-electric-field region) can occur owing to the decreased Schottky barrier width.

The rectifying behavior of the selector device is advantageous in terms of the crossbar array size. Figure 6a shows the half-bias scheme in the crossbar array. When a blue cell is selected, the red and green cells are half-biased and reversely half-biased, respectively. No bias was applied to the yellow cells. In the case of devices without rectifying behavior, the sneak path current occurred through the reverse half-biased cells (green cells). This sneak path current can become a significant limitation when the crossbar array size is increased. Based on Equation (Equation 6), which shows the normalized read-out margin ($\Delta V_{out}$/V${}_{pu}$) from Kirchhoff’s law, we calculated the maximum number of word lines (*N*) with at least a 10% readout margin.
(6)$$\begin{array}{c} \frac{\Delta V_{out}}{V_{pu}}=\frac{R_{pu}}{\left[R_{LRS}^{Select}\|\left(\frac{2R_{LRS}^{Sneak}}{N-1}+\frac{R_{LRS}^{Sneak}}{{\left(N-1\right)}^{2}}\right)\right]+R_{pu}}\\ -\frac{R_{pu}}{\left[R_{HRS}^{Select}\|\left(\frac{2R_{LRS}^{Sneak}}{N-1}+\frac{R_{LRS}^{Sneak}}{{\left(N-1\right)}^{2}}\right)\right]+R_{pu}} \end{array}$$
where *N*, *R*${}_{select}$, *R*${}_{sneak}$, and *R*${}_{pu}$ represent the maximum number of word lines, the parameters of the selected cell resistance, the sneak path resistance, and the voltage across the pull-up resistor, respectively, [23]. We determined *R*${}_{selected}$ to be the resistance of the blue cell, which is the selected cell. Therefore, the value of *R*${}_{selected}$ was selected as the resistance of the LRS: 1.02 MΩ (for HRS: 1.25 MΩ) at V${}_{read}$ (2 V). Secondly, *R*${}_{sneak}$ is the resistance of unselected cells, which can cause a sneak current. The sneak current can flow through three unselected cells (red, green, and yellow cells of Figure 6a) when the unselected cells are in the LRS. Therefore, the resistance values of unselected cells are 144.38 MΩ (red cells) and 73.48 GΩ (green cells) at half-read voltage (−1 V). In addition, we considered that zero voltage is applied to the yellow cells because the influence of the yellow cells can be ignored in very large *N*. Lastly, for *R*${}_{pu}$, the LRS resistance value of *R*${}_{Selected}$ was used to distinguish the resistance state of the selected cell. The defined parameters are summarized in Table 1.

Figure 6b shows the read-out margins by N for various cases based on the half-bias scheme. For the 10 μA current compliance case (blue line), N (number of word lines) increases from 13 to 46 compared to the case without a selector device (black line). In addition, a 10${}^{5}$-times higher N was achieved in the 2 μA current compliance case (red line).

## 4. Conclusions

In summary, an Ag-filament-based transparent threshold switch was successfully developed as a selector device. The device exhibited approximately 75% transmittance in the visible light region. To analyze the operation mechanism, the current compliance dependence was investigated, which exhibited not only a memory switch, but also a threshold switch. In addition, even though the threshold switches were the same (at 10 and 2 μA current compliance), a more obvious rectifying behavior was observed at 2 μA current compliance. Based on the analyses of the conduction mechanism, it was demonstrated that the rectifying behavior resulted from the Schottky barrier at the interface between the ITO and WO${}_{3}$ layers. Moreover, at 10 μA current compliance, a thicker filament, which leads to a narrower Schottky barrier width, was formed. Thus, direct tunneling was observed only for the 10 μA current compliance. Based on the achieved results (especially clear rectifying behavior at 2 μA current compliance), the maximum size of the crossbar array was significantly enlarged from 13 to 10${}^{5}$.

## Figures and Tables

**Figure 1 micromachines-13-01874-f001:**
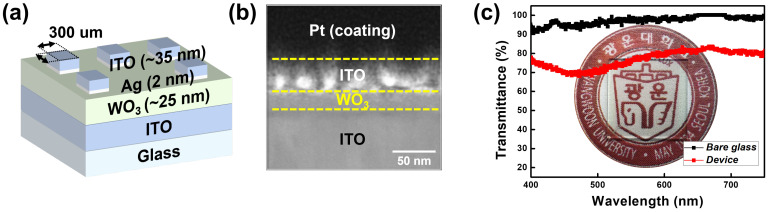
(**a**) Simple illustration of the developed Ag-based transparent threshold switch. (**b**) Cross-sectional TEM image of the developed device. (**c**) Optical transmittance of the device with photo image.

**Figure 2 micromachines-13-01874-f002:**
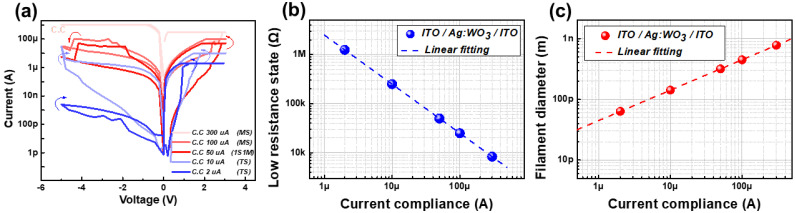
(**a**) Current versus voltage (I–V) curves at different current compliance values from 2 to 300 μA. Various characteristics such as memory switching (MS), one selector with one memory (1S1M), and a threshold switch (TS) are dependent on the current compliance. (**b**) Resistance in the LRS at various current compliance values. (**c**) Derived filament size (diameter) with the current compliance from 2 to 300 μA.

**Figure 3 micromachines-13-01874-f003:**
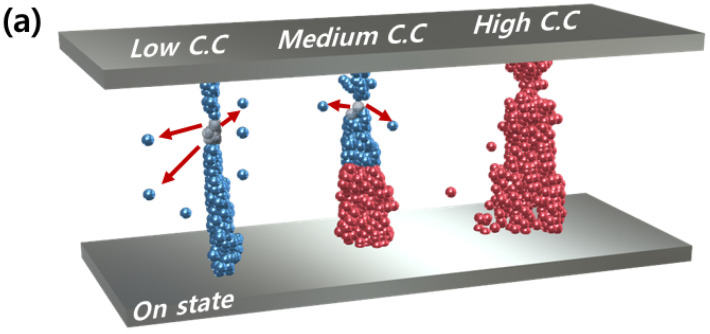
Operation schematics of (**a**) on and (**b**) off state at low, medium, and high current compliance. The thick and stable filament is represented as a red filament, while the thin and unstable filament is illustrated as a blue filament. At medium current compliance, both filaments coexist, which leads to the 1S1M characteristic. The unstable filament exhibits spontaneous rupture, which is illustrated by red arrows.

**Figure 4 micromachines-13-01874-f004:**
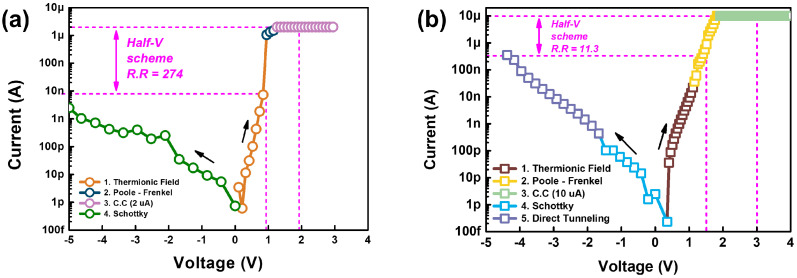
I–V curves of threshold switches at (**a**) 2 μA and (**b**) 10 μA current compliance. To analyze the conduction mechanism, the I–V curves are fitted by various conduction mechanisms. In the positive-bias region, the same conduction mechanisms such as thermionic field emission with Poole–Frenkel conduction are observed. In the negative-bias region, only the Schottky emission is observed at the 2 μA current compliance, while the Schottky emission with direct tunneling is observed at the 10 μA current compliance.

**Figure 5 micromachines-13-01874-f005:**
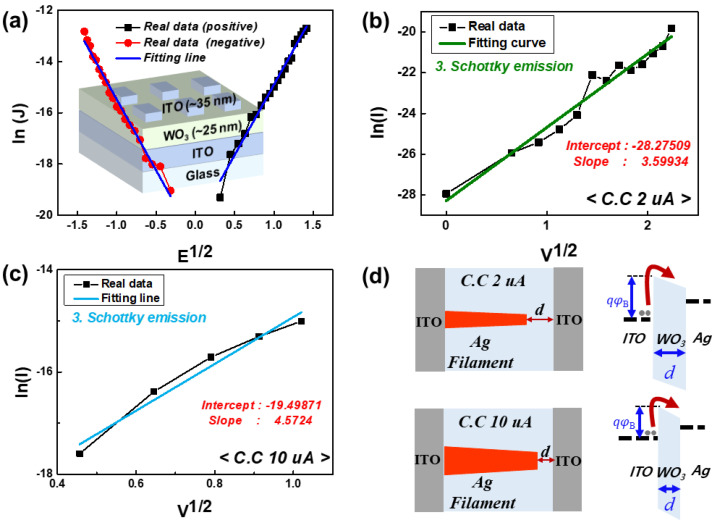
(**a**) Fit I–V curve of the symmetric structured device to demonstrate the rectifying behavior. The inset shows the structure of the fabricated device. The curve is well fit by the Schottky emission in both the positive- and negative-bias region. This means that the Schottky barrier is formed at the interface between the ITO and WO${}_{3}$ layers. For (**b**) 2 μA and (**c**) 10 μA current compliance, intercepts and slopes are extracted, from which the Schottky barrier height and width can be derived using Equations (4) and (5). (**d**) From the results, the operation mechanisms of the 2 μA and 10 μA cases can be depicted by the Schottky barrier height and width d.

**Figure 6 micromachines-13-01874-f006:**
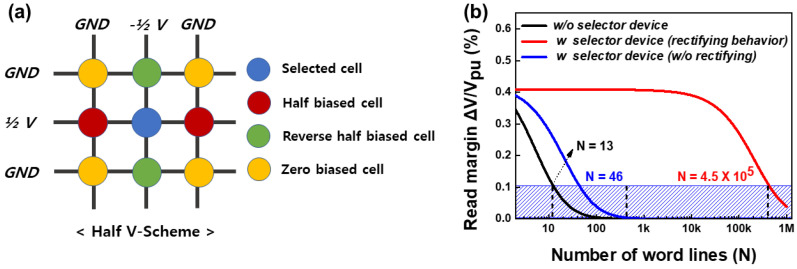
(**a**) Schematic of the half-bias scheme in the crossbar array. (**b**) Calculated read-out margins for various cases such as without the selector device, with the selector device, and with the rectifying selector device. The rectifying selector device exhibited an array size of more than 10${}^{5}$× 10${}^{5}$.

**Table 1 micromachines-13-01874-t001:** Defined parameters in various cases.

	w Rectifying Characteristics	w/o Rectifying Characteristics
*R* ${}_{LRS}^{selected}$	1.02 MΩ	1.02 MΩ
*R* ${}_{HRS}^{selected}$	1.25 MΩ	1.25 MΩ
*R* ${}_{LRS@positive}^{sneak}$	144.38 MΩ	144.38 MΩ
*R* ${}_{LRS@negative}^{sneak}$	73.48 GΩ	144.38 MΩ

## Data Availability

The datasets used and/or analyzed during the current study are available from the corresponding author upon reasonable request.

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
