# Peer review of "Ag-Ion-Based Transparent Threshold Switching Selector with Filament-Size-Dependent Rectifying Behavior"

_micromachines, 2022, doi:10.3390/mi13111874_

Round 1

Reviewer 1 Report

Seo J. et al., proposed low CC operation to produce threshold devices that can be exploited as a selector. The authors provided good explanation on the conduction mechanism point of view. This manuscript could contribute to the advancement of ReRAM technologies and benefit its field; however, several problems are identified in the manuscript that should be addressed before the reviewer can recommend the publication of this manuscript, and they are listed below:

(1)Materials & Methods: The authors should clarify whether the 2nm Ag were patterned into 300x300um square or blanket? Lines 29-31 contradict the schematic Fig. 1a.

(2)I am not convinced the Ag layer is uniformly diffused in to the whole stack (Fig.2b, Line 34). It can be seen from the cross-sectional TEM image, the interface is not clear indicating the TEM sample preparation is not for high-resolution TEM investigation; if it is HRTEM we could see the 2nm Ag layer. Nevertheless, the reviewer understands if the authors have difficulty to get good sample preparation. Anyway, the reviewer is sure that the Ag element just dispersed to the entire sample due to the poor etching process during sample preparation (either with FIB or ion milling, depending which technique you used), similar dispersion issue can be seen when you stack Cu layer; this is because Ag and Cu do not have good adhesion on oxide. I suggest the EDS should be deleted from the manuscript.

(3)The authors claimed that the switching occurred due to the rupture/formation of Ag filament. First, this contradicts Fig.2b (EDS), if Ag diffuses throughout the WO layer then most likely homogeneous switching should occur instead of filamentary. Secondly, the cation source (Ag layer) is only 2nm, which may not sufficient to inject enough Ag cations into the 25 nm of WOx layer. Hence the authors should do these well-accepted 2 simple protocols: 1. To confirm whether the switching in this devices is based on filamentary or homogeneous, please measure the LRS current at different cell size, please follow the procedure as suggested in paper [1], and 2. If protocol 1 indicates it is filamentary then in order to confirm whether it is Ag filamentary, oxygen vacancies or mixture of various defects-based filament, the authors should conduct temperature coefficient resistance test, as suggested in paper [2]. Note that, In originated from ITO, or/and another defects could co-exist and contribute to the formation of hybrid filaments as well, as reported in paper [3].

(4)Please show the curve fitting of the conductions stated in Figs. 4(c) and (d).

(5)In line with comment No.4, I suggest the colouring in Figs. 4(c) and (d) can be applied in Figs (a) and (b) to concise the number of the displayed curves in Fig. 4.

(6)Fig. 2(a). In my experience, most reram devices that sweep below its switching current are indeed showing rectifying and threshold behaviour, however these behaviours tend to last only for few cycles. The authors should conduct a repeating sweep for each of this CC parameter to ensure these rectifying/threshold behaviours are not a temporary behaviour; I believe single-sweep mode (0V to -/+5V only, not need to include -/+5V to 0V sweep direction) for both forward and reverse biases for 50-100 cycles are sufficient.

(7)In line with comment No.6, hence, each CC parameter can have its own subfigure, where all the repeated sweeps are plotted together. Henceforth, the reader could see the stability of these devices.

(8)The authors should clarify how they determine the Rselect, Rsneak, and Rpu values; this is to ensure the general readers understand the concept behind this formula.

(9)In addition to the explanation given for Fig.3 (low CC) in the text, the authors should also add a brief explanation mentioning that in CC2uA case, a larger portion of the conduction channel (filament or homogeneous, whichever you identified) structure in CC2uA ruptured than that of the 10uA case, as indicated by the IVs shown in Fig.2(a) which has very low current during the negative bias.

(10)The authors should study and cite these papers to concise the literature review:

[1] Manipulated Transformation of Filamentary and Homogeneous Resistive Switching on ZnO Thin Film Memristor with Controllable Multistate (doi: 10.1021/am4007287 )

[2] Transformation of digital to analog switching in TaOx-based memristor device for neuromorphic applications (doi: 10.1063/5.0041808)

[3] Fast, Highly Flexible, and Transparent TaO x -Based Environmentally Robust Memristors for Wearable and Aerospace Applications (doi: 10.1021/acsaelm.0c00441)

[4] Switching Failure Mechanism in Zinc Peroxide-Based Programmable Metallization Cell (doi: 10.1186/s11671-018-2743-7)

Author Response

Thank you very much for your kind comments. The manuscripts were revised according to your comments. And the point-by-point response is attached here.

Reviewer 2 Report

I am pleased to see the work by Seo and colleagues. There has been recent interest in devices that can act as both a synapse and a neuron. This work demonstrates one such solution. I have no problems with the experimental data and also the qualitative model. I warn the editor that there is no direct experimental evidence to support the filament model, but there is indirect evidence in the form of rigorous fits to electrical data. I don't see a problem with such indirect evidence, as long as it works for the editor. I look forward to seeing the paper published.

Author Response

Thank you very much for your kind comment. The manuscript has been carefully revised for the readers.

Round 2

Reviewer 1 Report

The authors have addressed all comments very well. Now I can recommend the publication of this manuscript. Very good work.